# The Nitrogen Dynamics of Newly Developed Lignite-Based Controlled-Release Fertilisers in the Soil-Plant Cycle

**DOI:** 10.3390/plants11233288

**Published:** 2022-11-29

**Authors:** Gunaratnam Abhiram, Miles Grafton, Paramsothy Jeyakumar, Peter Bishop, Clive E. Davies, Murray McCurdy

**Affiliations:** 1Environmental Sciences, School of Agriculture & Environment, Massey University, Private Bag 11 222, Palmerston North 4442, New Zealand; 2Department of Export Agriculture, Faculty of Animal Science and Export Agriculture, Uva Wellassa University, Badulla 90000, Sri Lanka; 3School of Food and Advanced Technology, Massey University, Private Bag 11 222, Palmerston North 4442, New Zealand; 4Verum Group, Lower Hutt 5010, New Zealand; 5GNS Science, Lower Hutt 5010, New Zealand

**Keywords:** controlled-release fertiliser, Fe^2+^ application, lock-off nitrogen, nitrogen leaching, nitrous oxide emission, soil residual nitrogen

## Abstract

The effect of newly developed controlled-release fertilisers (CRFs); Epox5 and Ver-1 and two levels of Fe^2+^ applications (478 and 239 kg-FeSO_4_ ha^−1^) on controlling nitrogen (N) losses, were tested on ryegrass, in a climate-controlled lysimeter system. The Epox5 and Ver-1 effectively decreased the total N losses by 37 and 47%, respectively, compared to urea. Nitrous oxide (N_2_O) emissions by Ver-1 were comparable to urea. However, Epox5 showed significantly higher (*p* < 0.05) N_2_O emissions (0.5 kg-N ha^−1^), compared to other treatments, possibly due to the lock-off nitrogen in Epox5. The application of Fe^2+^ did not show a significant effect in controlling the N leaching loss and N_2_O emission. Therefore, a dissimilatory nitrate reduction and chemodenitrification pathways were not pronounced in this study. The total dry matter yield, N accumulation, N use efficiency and soil residual N were not significantly different among any N treatments. Nevertheless, the N accumulation of CRFs was lower in the first month, possibly due to the slow release of urea. The total root biomass was significantly (*p* < 0.05) lower for Epox5 (35%), compared to urea. The hierarchical clustering of all treatments revealed that Ver-1 outperformed other treatments, followed by Epox5. Further studies are merited to identify the potential of Fe^2+^ as a controlling agent for N losses.

## 1. Introduction

Pastoral agriculture is the backbone of New Zealand’s (NZ) economy and nearly 9 million hectares of land (40% of the total land area in NZ) are under pastoral farming [1]. To increase pasture productivity and quality, N fertiliser is applied mainly in the pastoral dairy farming sector. However, applications of readily available N fertilisers to pastoral land increase the nitrogen (N) losses through leaching (nitrate: NO_3_^−^-N) and gas emissions (N_2_, N_2_O, NH_3_). These losses of N in pastoral lands can result in the degradation of the water and air quality [2]. For example, Richards, et al. [3] reported that about 15% of New Zealanders are facing the risk of exposure to high nitrate levels in drinking water. As a result, several N management strategies are being used to mitigate the N losses such as split application, timing, precision placement of the fertiliser, growing fallow crops, riparian plantings, and carbon source management in the soil [4]. In general, these methods are time-consuming and labour-intensive [4]. Controlled-release fertilisers (CRFs) have been shown to be an effective strategy to mitigate N losses in many parts of the world [5,6]. Controlled-release fertilisers prolong the N release, allowing the plants to uptake more N than conventional N fertilisers [7]. Consequently, the losses to the environment decrease and the N utilisation efficiency (NUE) increases.

Further, the N transformation pathway in soil has direct links to soil mineralogy and biology. Several elements (e.g.,: C, Cu^2+^, H, P and Fe^2+^) in the soil have been shown to play a significant role in the N cycle [8,9]. Iron is one of the important elements in soil, which has been shown to influence the nitrogen cycle. The role of ferrous (Fe^2+^) and ferric (Fe^3+^) ions in N cycles are not yet fully understood [10] and is a topic currently under study. Iron’s effect on the N cycle is referred to as dissimilatory iron-reduction (DIR) where microbial and chemical reactions take part in the denitrification process. In a non-sulfidic environment, the dissimilatory iron-reducing bacteria (DIRB) reduce Fe^3+^ to Fe^2+^ by oxidizing the organic carbon [8,11]. The resulting Fe^2+^ ion is coupled to the nitrate leaching reduction by this DIRB. Whilst under anaerobic conditions, the process yields Fe^3+^ and reduced forms of N, predominantly N_2_ [8,12]. This DIR process is reported in many agricultural soils when anoxic conditions are present [13,14].

In a previous study, it was found that a high concentration of Fe^2+^ in subsoil (iron-rich sand) under anaerobic conditions decreased nitrate in leachate and nitrous oxide emissions possibly due to dissimilatory nitrate reduction (DNR) [4]. The present case study investigates the impact of the surface application of Fe^2+^ on the nitrate reduction. In addition, the application of two new CRF formulations (Epox5 and Ver-1) in controlling N losses on ryegrass under simulated climate conditions are investigated. However, the effect of the Fe^2+^ application on the nitrate reduction pathway was not examined in this study.

## 2. Results and Discussion

### 2.1. Climate Data

The climate emulated in the lysimeter had 28 rainfall events and 26 of them created drainage (Figure 1). The cumulative monthly rainfalls were 380, 409 and 443 mm, respectively, for the first three months, and the corresponding cumulative drainage values were 199, 189 and 147 mm. The air temperature in the lysimeter caps fluctuated between 7 and 16 °C, whilst the soil temperature varied between 10 and 17 °C (Figure 1).

### 2.2. Nitrate (NO_3_^−^ -N) and Ammonium (NH_4_^+^ -N) Leaching Losses

The nitrate leaching losses were higher in the first two months, compared to the third month (Figure 2a). The cumulative nitrate leaching losses in the first month were 34, 25, 30, 45 and 34 kg-N ha^−1^ for urea, Epox5, Ver-1, Fe 7.5 and Fe 15, respectively (Figure 2a). The corresponding values for the second month were 65, 44, 37, 76 and 69 kg-N ha^−1^ and they remained nearly constant in the third month. This suggests that not enough nitrate was left in the soil for the leaching losses in the third month. The control and NC treatments showed lower nitrate leaching losses throughout the experimental period. The peaks of nitrate loss for most treatments (except the control and NC treatments) were observed after 15, 23 and 48 days of the fertiliser application which followed the high rainfall events.

Ammonium leaching occurred right through the experimental period with intermittent leaching peaks (Figure 2b). The highest ammonium losses were recorded from 48 to 68 days of leaching events for all treatments. This is likely due to the positive charge of ammonium ions so they moved slower in the soil profile than nitrate ions as the soil has a negative surface charge which attracts positive ammonium ions that slow down the movement.

The total nitrate leaching was significantly (*p* < 0.05) lower in the control and NC treatments, compared to all other N treatments (Figure 3a). Among the N treatments, the total nitrate leaching was significantly (*p* < 0.05) reduced by 37 and 47% in Ver-1 and Epox5-treated soils, respectively, relative to the urea treatment. A 16% nominal reduction in nitrate loss was observed in the Ver-1 treated soil, in comparison to Epox5, but it was not statistically significant. The total ammonium losses were much less than that of nitrate, they ranged from 1.6 to 2.6 kg-N ha^−1^, and were not significantly different between treatments (Figure 3a).

The total N leaching loss is the sum of both nitrate and ammonium losses (Figure 3b). The soil treated with Ver-1 showed a significantly (*p* < 0.05) lower total N loss, compared to other N treatments, except for Epox5. Nitrate leaching is the predominant pathway for N losses through an agriculture system [15]. The results of this study showed that both CRFs (Epox5 and Ver-1) were effective in controlling the nitrate-N leaching loss by 37 and 47%, respectively, compared to urea only. This shows that both CRFs slowly release N and prolong the N release duration. These nitrate leaching loss reduction values were higher than the average leaching reduction value of 10% reported for polymer-coated CRFs applied in many grasslands [16]. Both chemodenitrification and DNR pathways involve denitrifying the nitrate [17], which could have decreased the nitrate in leachate. However, the iron (FeSO_4_) application with urea did not reduce the nitrate and ammonium leaching losses, which implies that both of these pathways were not prominent in this study. The Ver-1 significantly (*p* < 0.05) lowered the total mineral N losses, compared to Ver-c. This shows that this effect was due to the formulation of CRF (Ver-1), not the mere interactions between lignite and urea (Ver-c). The application of lignite with urea (Ver-c) did not significantly decrease the N leaching losses and therefore, the lignite application may not be an effective leaching control measure. The results support a previous study conducted in New Zealand which reported that lignite was ineffective in controlling the nitrate leaching from urea and biosolids [18].

### 2.3. Nitrous Oxide Emission

The N_2_O emission of the NC treatment was the lowest among all of the treatments. However, it was higher than other treatments on the day one gas collection (Figure 4a). This observation may be due to the absence of ryegrass which led to the non-utilisation of the soil residual mineral N. This condition probably favoured the denitrification process and released higher N_2_O, compared to other treatments. However, in the following days, the N_2_O emissions were close to zero in the NC treatment. A similar observation was reported by Hénault, et al. [19] in a field study where the N_2_O emissions were high in the initial days and declined in subsequent days in bare soil.

In general, the peak N_2_O emissions were reached 15–25 days after the fertiliser application, which is consistent with a previous finding [4]. Forty-five days later, the emission was close to zero for all treatments, except for Epox5 (Figure 4a). The zero-emission demonstrated that the mineral N was unavailable on the topsoil layers for the denitrification process. In contrast to other treatments, the Epox5 treatment continued to release N_2_O up to the last day (88th day) (Figure 4a). At the end of the experiment, Epox5 fertiliser coating shells were collected and analysed for the residual N. It was found that 9.7 ± 0.3 kg-N ha^−1^ amount of unreleased N remained in the coating, which supports the reason for the continuous N_2_O release measured.

The total N_2_O emissions were significantly (*p* < 0.05) lower for the NC treatment, compared to the Epox5 and Fe 7.5 treatments, but not significantly different from other treatments (Figure 4b). The Epoxy5 exhibited the highest N_2_O emissions of 0.5 kg-N ha^−1^, which was 0.35 kg-N ha^−1^ (233%) higher than the urea treatment. This difference might be explained through the slow release of N by Epox5, which continued to supply N available for the denitrification process which has increased the emission. Although this is a new finding for ryegrass, similarly higher N_2_O emissions from sulphur-polymer-coated urea, when compared with granular urea, was reported on sugarcane in a Brazilian study [20]. The N_2_O emissions of Ver-1 were not significantly different from any other treatments, except Epox5 (Figure 4b). The lack of significant N_2_O emission differences between the urea treatments (except Epox5) is possibly due to the high leaching conditions which existed during this study as the climate model had high rainfall events and the lysimeters have a sandy subsoil.

The Fe^2+^ acts as an electron donor for the denitrification of nitrate and enhances the N_2_O emissions in the chemodenitrification process [17]. However, the application of FeSO_4_ did not significantly increase the nitrous oxide emissions in this study, which suggests that chemodenitrification was not pronounced. Meanwhile, iron treatments did not significantly decrease the N_2_O emissions, compared to urea, which shows DNR was also not prominent in this case study.

### 2.4. Residual Soil Nitrogen

The residual mineral nitrogen (nitrate and ammonium) in the soil was measured at 5, 10, 20, 30 and 40 cm depths (Table 1). The total mineral nitrogen was calculated by summing the nitrate and ammonium content at the respective depths. The residual nitrate, ammonium and total nitrogen content were not significantly different between any treatments at any depth (Table 1). The total residual nitrate and ammonium were between 53.5–82.5 and 6.5–14.4 kg-N ha^−1^, respectively. The residual nitrate made up a larger portion of the residual N, compared to ammonium. The residual nitrate decreased with depth for all treatments and the major portion resided in the first 10 cm depth of the soil profile (Table 1). In this study, the top 10 cm of the soil profile is composed of topsoil that has a higher anion storage capacity than sandy subsoil. Similarly, residual ammonium decreased with depth, but the decrease was not as prominent as nitrate (Table 1).

### 2.5. Dry Matter Yield

The control treatment showed a significantly (*p* < 0.05) lower DM yield in the first month than other treatments, except for Fe 15 (Figure 5a). The DM yield of the second and third months was not significantly different between N treatments (Figure 5b). The total DM yield was significantly (*p* < 0.05) lower for the control, compared to other treatments. The total DM grass yields were 672, 1958, 2083, 1798, 1543, 1614 and 1992 kg ha^−1^ for the control, urea, Fe 7.5, Fe 15, Epox5, Ver-C and Ver-1 treatments, respectively (Figure 5d). The stubble DM was not significantly different between treatments (Figure 5e). Although the above-ground total biomass (AGTBM) was significantly (*p* < 0.05) higher for all N treatments, relative to the control, however, there were no significant differences among them (Figure 5f).

In terms of the total grass DM, the application of the Epox5 treatment reduced the total grass DM by 21% relative to the lysimeters treated with urea only, however, this increase was not significantly different. Similarly, the application of both iron treatments (Fe 7.5 & Fe 15) did not result in a significant increase in the total grass DM, compared to urea. The iron application at these levels (7.5 & 15 mg FeSO_4_ kg^−1^ of soil) has not significantly impaired the DM yield. These results contrast with results reported by Wong and Bradshaw [21]. The authors reported that the application of Fe at 15 mg-FeSO_4_ L^−1^ of a solution level resulted in a significant reduction of the ryegrass DM yield (24%) in a hydroponic study.

### 2.6. N Accumulation and N Utilisation Efficiencies

The herbage N recovery by ryegrass cuttings of each month and stubble cuttings at the end of the experiment are tabulated in Table 2. The total herbage N was calculated by summing the N uptake of the monthly grass cuttings.

The control treatment showed a significantly (*p* < 0.05) lower herbage N, compared to other treatments in all three cuttings (Table 2). The herbage N in the first grass cutting was significantly (*p* < 0.05) higher for the Ver-1 treatment, compared to the control, Fe 15 and Epox5 treatments, but not significantly different from other treatments (Table 2). In the second cutting, the Fe 15 treatment showed a significantly (*p* < 0.05) higher herbage N, compared to the control, Epox5 and Ver-c, however, did not significantly differ from other treatments (Table 2). The third cutting and the total herbage N were significantly lowest (*p* < 0.05) for the control, whereas other treatments did not show significant variations from each other (Table 2). The N content of the stubble was significantly (*p* < 0.05) lower for the control, compared to urea, Epox5, Ver-c and Ver-1, but did not significantly differ from both iron treatments. Both AUE and ARE were not significantly (*p* < 0.05) different between any treatments. All of the treatments showed very low ARE values ranging between 0.21 and 0.31. These observed lower ARE values were possibly due to the high N-leaching losses.

The Epox5 provided a significantly (*p* < 0.05) lower nitrogen recovery than Ver-1 in the first month but did not significantly differ in the last two months. Both CRFs did not significantly increase the AUE and ARE, compared to the urea treatment (Table 2). The total herbage N decreased by 27% in comparison to Ver-1, but this reduction was insignificant. Other studies have reported that the CRF application lowered the herbage N recovery [6,22]. Both iron treatments did not significantly alter the N recovery in relation to the urea treatment.

### 2.7. Root Dry Matter Distribution

The root DM was measured at five different depths; 0–5, 5–10, 10–20, 20–30, and 30–40 cm. The root DM did not significantly differ among treatments along the soil profile except for a few treatments at a 0–5 cm depth (Figure 6a). Only urea and Ver-1 showed a significantly (*p* < 0.05) higher root DM, compared to the Epox5 at 0–5 cm depth. The percentages of the root DM distribution were 60–65%, 14–18%, 12–14%, 5–8% and 3–4% for incremental depths from the top of the soil. The total root DM (TRDM) was significantly (*p* < 0.05) higher for the urea and Ver-1 treatments, relative to Epox5, but there was no significant difference among other treatments (Figure 6b). This lower TRDM for Epox5 can be explained using root plasticity. It is documented that plant root biomass and extensions are inhibited by a high N application and vice versa [23,24], and is the case with perennial ryegrass [24]. A continuous supply of nitrogen by Epox5 could have decreased the demand for nitrogen and impaired the root growth and extension, compared to other treatments.

### 2.8. Nitrogen Budget

The mineralisation during the experimental period was calculated using the N balance model. The N mineralisation calculated for the control (with grass) treatment was 84.6 kg-N ha^−1^ and was assumed as the N mineralisation value for all other treatments (Table 3). All of the treatments showed a surplus N balance, in which the highest value was observed in the Epox5 treatment (Table 3). The surplus N balance in this study could be attributed to the following reasons; unmeasured N components, such as root N, ammonia volatilisation, immobilisation and denitrification. In addition, using the N mineralisation value calculated for the control could be an overestimation for all other treatments. The rate of mineralisation of the fertilised soil is lower than that of the unfertilised soil in cultivated land use (control) [25].

### 2.9. Hierarchical Clustering of the Fertiliser Treatments

Hierarchical clustering was used to identify the best and closely related fertiliser treatment/s, based on the overall performances of the nitrate leaching loss reduction, total N leaching loss reduction, N_2_O emission reduction, total DM yield, herbage N and soil residual N. The nitrate leaching loss reduction, total N leaching loss reduction and N_2_O emission reduction were the distinct variables between treatments that mainly influenced the cluster method. Other variables showed a low variation between treatments, and therefore, their influences in the clustering were not substantial (Figure 7). Ver-1 was the best treatment, based on the overall performances, which showed the highest values for nitrate and the total N leaching loss reductions, and the second-highest N_2_O emission reduction. Both CRFs (Ver-1 and Epox5) showed better performances than other treatments and their performances were more closely associated with reducing nitrate and N leaching losses (Figure 7). For example, nitrate leaching loss reductions were 84 and 100% for Epox5 and Ver-1 treatments, respectively. The urea treatment was grouped separately from any other fertiliser treatments. Fe 15 and Ver-c treatments were more associated with each other since both treatments showed a similar N loss profile. For example, nitrate leaching loss reductions were 53 and 46% for Fe 15 and Ver-c, respectively.

### 2.10. Importance of the Study for Pastoral Agriculture

Nitrogen losses from pastoral land have been a big concern in New Zealand. In New Zealand, conventional fertilisers, such as urea are used in the pastoral field and they release nutrients to the soil in a rapid manner. CRFs are a proven strategy for controlling N losses. However, CRFs are not widely used in pastoral land and only limited knowledge is available on how CRFs behave in the pasture. This case study showed that N leaching losses can be mitigated by using CRFs Epox5 and Ver-1. One of the aims of using CRFs is to minimise N_2_O emissions. However, the Epox5 treatment significantly (*p* < 0.05) increased N_2_O emissions, three times higher than urea. The analysis of the coating membrane at the end of the experiment revealed that unreleased (lock-off) urea was found on the coating’s inner surface, which supports the above observation. This suggests that controlling N leaching could increase the likelihood of N_2_O emissions. Therefore, Epox5 fertiliser is more suitable for the regions and/or seasons receiving lower rainfall, to minimise denitrification in anaerobic conditions which are linked to N_2_O emissions.

In general, the newly developed CRF formulations controlled the leaching losses without a significant loss of the DM yield, compared to the urea treatment. The insignificant effect of these formulations on the DM yield was a positive response, when compared to the 50% DM yield reduction reported for CRFs on various grasslands [16]. Despite both CRFs significantly (*p* < 0.05) decreasing the N leaching losses, it was not used by the ryegrass. Both CRFs showed an insignificant difference (*p* < 0.05) in the herbage N recovery, compared to urea and therefore, leaching control does not always improve the nitrogen utilisation efficiencies. This observation could be due to the mismatches between the N demand of ryegrass and N supply by the CRFs.

## 3. Materials and Methods

### 3.1. Experimental Design

A specially designed lysimeter system which can emulate a selected climate model was used in this study, as described in Gunaratnam [4] and McCurdy, et al. [26]. This innovative lysimeter system consists of two components; capped lysimeters and a climate-control unit (CCU). There were 40 lysimeters 45 cm in height and 20 cm in diameter which were capped to provide a confined space for controlling the climate (Figure 8). A CCU was coupled with the lysimeters to emulate a climate model. Lysimeters were repacked with Manawatu fine sandy loam soil for the top 0–10 cm (topsoil) and washed builders’ sand for the subsoil (10–30 cm). Topsoil was collected from a dairy farm at Massey University, New Zealand and washed builders’ sand was purchased from Mitre 10, Lower Hutt, New Zealand. A large portion of the subsoil was composed of sand to increase the drainage and the amount of N in leachate. A detailed description of the soil repacking method can be found in Gunaratnam, et al. [27]. In summary, the top and subsoil were packed at an incremental height of 5 cm and a fibreglass wick was attached at the bottom boundary of the soil matrix to provide capillary suction.

Perennial ryegrass (*Lolium perenne* L.) swards were transplanted into the lysimeters 3-weeks after planting at 10 seedlings per lysimeter. The ryegrass was grown in the lysimeters for 3 months, under LED grow lights in a grow tent, until they were ready to use for experiments. During the ryegrass growth period, the soil moisture was maintained at nearly 70% of the field capacity. Ryegrass swards were clipped at a 5 cm height to mimic the natural grazing and induce the tiller formation at the end of each month. Nitrophoska Extra’ (Ravensdown) fertiliser was applied after the first clipping at the rate of 50 kg-N ha^−1^. This fertiliser contains N, P, K, S, Mg and Ca in 12:5.2:14:8:1.2:3.8 ratios (and trace elements). Prior to transplanting the swards, several pore volumes of water (4 to 5) were dripped on the soil column to leach the residual nutrient in the soil. At the end of three months, the transplanted ryegrass tillers were clipped and the soil moisture level was brought to 100% on the starting day of the experiment, to maintain a uniform moisture content for all experimental units, and facilitate the water balance calculations.

### 3.2. Treatments and Climate Model

The experiment was conducted in climate-controlled lysimeters (Figure 8) with eight treatments. These treatments consist of a negative control (NC), control, urea only (200 kg-N ha^−1^), urea + 239 kg FeSO_4_ ha^−1^ (Fe 7.5), urea + 478 kg FeSO_4_ ha^−1^ (Fe 15), Epox5, Ver-c and Ver-1. The NC treatment was a bare soil-sand matrix without ryegrass, but the control was planted with ryegrass, and both the NC and control treatments did not receive N fertilisers. The application of 239 and 478 kg-FeSO_4_ ha^−1^ was equivalent to 7.5 and 15 mg FeSO_4_ kg^−1^ of soil, respectively. Epox5, a lignite-epoxy coated urea and Ver-1, urea-impregnated lignite, two newly developed different types of controlled-release fertilisers were tested. The treatment Ver-c was a blend of 1200 kg-lignite ha^−1^ (particle size < 1 mm) and a granular urea (compositions and rates were equivalent to Ver-1). This treatment Ver-c was used to test the actual effect of the control release fertiliser (Ver-1) rather than the interaction effect of urea and lignite. Fertilisers, FeSO_4_ and lignite were applied as single surface applications at the start of the experiment. All treatments were replicated five times, and the experimental setup was arranged in a completely randomised design (CRD). The detailed preparation process and characterisation of Epox5 can be found in Gunaratnam [4]. It is urea coated with five layers of lignite-epoxy composite. Ver-1 is a controlled-release lignite and N fertiliser developed by Verum Group, Lower Hutt, New Zealand and the method of preparation is embargoed.

A climate model from New Zealand’s Taranaki region was used in this study. Taranaki has a high annual rainfall pattern and is located on the west coast of the North Island. The combination of high rainfall and high dairy cow numbers in this region increases the susceptibility of nitrate leaching, compared to other parts of New Zealand. In particular, the spring season of 2013 climate model of this region was selected because it is within the first standard deviation of the 10-year average. A climate station named Stratford EWS (Agent No. 23872, Lat. −39.33726°, Long. 174.30487°, Elev. 300 m) in Taranaki was used for this study and the climate variables were retrieved from the National Institute of Water and Atmospheric Research’s (NIWA) climate database. Daily rainfall values and weekly average values of temperature, relative humidity, and photosynthetically active radiation (PAR) hours were used in this case study. This was accomplished to minimise the practical difficulties of adopting daily values.

### 3.3. Leachate Analysis

Leachates were collected after every drainage event in a catch can and a subsample of 50 mL was filtered using 0.45 μm cellulose acetate filter paper. All filtered samples were stored at −4 °C for about two weeks until used for the analysis. Leachate subsamples were first thawed at room temperature before the analysis. The mineral N ions (NO_3_^−^ and NH_4_^+^) were measured using a Technicon Autoanalyzer, Series 2 [28].

### 3.4. Lock-Off Nitrogen in CRFs

The Epox5 granule shells were collected at the end of the experiment to measure the lock-off (unreleased) N. The shells were visually tested for unreleased urea and ground to powder. The powdered fertiliser shells were washed with deionised water several times to remove urea from the coating membrane and filtered using 45- micron filter paper. The filtrate was measured for the total N using the micro-Kjeldahl digestion method [28]. The lock-off N in the Ver-1 fertiliser was not measured since it was in powder form which made it difficult to completely retrieve from the soil.

### 3.5. Nitrous Oxide Measurement

The lysimeter cap was used for the gas collection which is similar to the static chamber method described by Collier, et al. [29]. The method used in this study is described in Gunaratnam [4]. To summarise, the valve in the air inlet port was closed and the air exhaust port was sealed with a rubber bung to collect the gas in an air-sealed space. Gas subsamples (35 mL) were collected at 0, 1 and 1.5 h from setting up the system for the gas collection. A 60 mL syringe was used for the gas collection and the subsamples were collected after homogenising the air in the space by depressing the piston four times. The collected subsamples were transferred to a pre-evacuated 12 mL vial and measured for N_2_O using a gas chromatograph (GC-2010, Shimadzu, Tokyo, Japan) coupled with an electron capture detector. The N_2_O flux was calculated using the following Equation (1);
(1)$$F=\frac{d_{C}}{d_{t}}\times \rho \times H$$
where F is the nitrous oxide flux (µg-N m^−2^ h ^−1^), d_C_/d_t_ is the nitrous oxide flux increment in the headspace in one hour (µL L^−1^), ρ is the nitrous oxide gas density (g m^−3^) and H is the lysimeter cap-height (m).

This daily flux was integrated into the total experimental period to obtain the total nitrous oxide emission.

### 3.6. Soil Residual Nitrogen

At the end of the experiment, the soil matrix was retrieved into five blocks using a specially designed lysimeter soil retriever [4]. The topsoil was sectioned into two 5 cm blocks while the subsoil was split into three 10 cm blocks. A 200 g amount of soil subsample was collected from each block and stored at −4 °C (not more than 3 weeks) until it was analysed. Once thawed, the subsample at room temperature, the soil was manually homogenised. Then, 3 g of soil was treated with a 2 M KCl solution and shaken in an end-over-end shaker for 1 h. The solution was centrifuged at 9000 rpm (107,100× *g*) for 10 min and the supernatant was filtered using a Whatman No. 42 filter paper. The mineral N of samples was analysed using a Technicon Autoanalyzer, Series 2 [28].

### 3.7. Grass Analysis

#### 3.7.1. Grass and Stubble Harvest

Ryegrass swards were clipped at 5 cm height every four weeks for a duration of three months. Stubbles were only clipped at the end of the third month at the ground height. The dry matter (DM) weights of grass and stubble were obtained after oven-drying the samples at 65 °C until a constant weight was obtained. The grass and stubble DM were added to obtain the DM of above-ground total biomass (AGTB).

#### 3.7.2. N Accumulation in Plant

The oven-dried ground grass and stubble sub-sample (1 g) was digested separately using the micro-Kjeldahl digestion method. The total N concentration was analysed using an auto analyser [28]. The total N uptake was calculated from the total DM yield of grass and stubble, and the N concentration.

### 3.8. Root Analysis

Once the soil blocks were retrieved from the lysimeters with the help of a specially designed lysimeter soil retriever [4], they were air-dried. The clods in the air-dried soil were rolled broken and sieved through a 1 mm sieve to separate the roots from the soil. Then the roots in each block were washed and the dry weights were obtained after oven-drying the root samples at 65 °C, until a constant weight was achieved.

### 3.9. N Use Efficiency

The agronomic use efficiency (AUE) and apparent recovery efficiency (ARE) were calculated using the following equations.
(2)$$\mathrm{AUE}=\frac{\mathrm{DM}\;\mathrm{yield}\;\mathrm{of}\;N\;\mathrm{treatment}-\mathrm{DM}\;\mathrm{yield}\;\mathrm{of}\;\mathrm{control}}{\mathrm{Applied}\;N}$$
(3)$$\mathrm{ARE}=\frac{\mathrm{HN}\;\mathrm{uptake}\;\mathrm{of}\;N\;\mathrm{treatment}-\;\mathrm{HN}\;\mathrm{uptake}\;\mathrm{of}\;\mathrm{control}\;\mathrm{treatment}}{\mathrm{Applied}\;N}$$
where DM is dry matter, HN is the herbage N uptake, and applied N is 200 kg N ha^−1^.

### 3.10. Nitrogen Budget

The N balance for all treatments was calculated using the following equation;
(4)$$N\;\mathrm{Balance}={\;F}_{N}+M_{N}+{\mathrm{IM}}_{N}-{\mathrm{FM}}_{N}-H_{N}-L_{N}-G_{N}$$
where F_N_ is the fertiliser N, M_N_ is the soil mineralised N, IM_N_ is the initial soil mineral N at the start of the experiment, FM_N_ is the final soil mineral at the end of the experiment, H_N_ is the N uptake by herbage, L_N_ is N lost via leaching and G_N_ is N lost in nitrous oxide form.

The prime interest of this case study is to minimise the N leaching losses and nitrous oxide (N_2_O) emissions via developing CRFs and therefore, other N gas species were not measured or considered in the N balance calculation used.

The net soil N mineralisation (M_N_) during 3 months was calculated from the control treatment using the following equation.
(5)$$\mathrm{Net}\;\mathrm{soil}\;N\;\mathrm{mineralisation}\;\left(M_{N}\right)={\;H}_{N}+L_{N}+G_{N}+{\mathrm{FM}}_{N}-{\mathrm{IM}}_{N}$$
where M_N_ is the soil mineralised N, H_N_ is the N uptake by herbage, L_N_ is N lost via leaching, G_N_ is N lost in nitrous oxide form, FM_N_ is the final soil mineral at the end of the experiment and IM_N_ is the initial soil mineral N at the start of the experiment.

### 3.11. Data Analysis

A statistical analysis was performed using Minitab 18 (Minitab, State College, PA, USA). The treatment effect on the different parameters was analysed using the one-way ANOVA test, followed by the Tukey honest significant difference (HSD), post hoc test for the mean separation at a 5% significance level. The data are presented as mean ± standard deviation of five replicates.

The hierarchical clustering was performed to group the fertiliser treatments, based on the following parameters; nitrate leaching reduction, total N leaching reduction, N_2_O emission reduction, total DM yield, herbage N uptake and soil residual N and was conducted using Origin 2019 Pro (Origin Lab Corporation, Northampton, MA, USA). The measured result of each parameter was normalised by calculating the percentage to avoid bias. In this way, the treatment which had the maximum value for a parameter received 100%, and other treatments were reported as a percentage proportionally.

## 4. Conclusions

This lysimeter study tested different types of controlled-release fertilisers and different iron application levels for controlling N losses. The following hypotheses were tested; (a) the performances of new CRF formulations were better than urea, and (b) the iron application (with urea) under aerobic conditions promotes a dissimilatory nitrate reduction. Both CRFs decreased the N leaching losses, whilst Epox5 increased N_2_O emissions (233%), compared to the urea treatment. These CRFs did not significantly increase the DM yield and the N utilisation efficiency (NUE), compared to urea. The overall performance analysis using the hierarchical clustering method revealed that Ver-1 performed better than other treatments, followed by Epox5. The better performance of Ver-1 on controlling N losses than Ver-c implied that the fertiliser formulation impacts the performance, not the interaction of urea with lignite. Iron (Fe^2+^) treatments were ineffective in controlling the N losses as they showed similar N loss profiles to urea. A pronounced DIR pathway in the iron (Fe^2+^) treatment could be indicated by a significance in all forms of N losses from soil, compared to the urea treatment. However, insignificant results between the iron (Fe^2+^) and urea treatments revealed that the application of Fe^2+^ with urea has not altered the dissimilatory nitrate reduction and chemodenitrification pathways under aerobic conditions. Further studies are recommended to test the effect of Fe^2+^ on promoting the aforementioned pathways on different types of soil.

## Figures and Tables

**Figure 1 plants-11-03288-f001:**
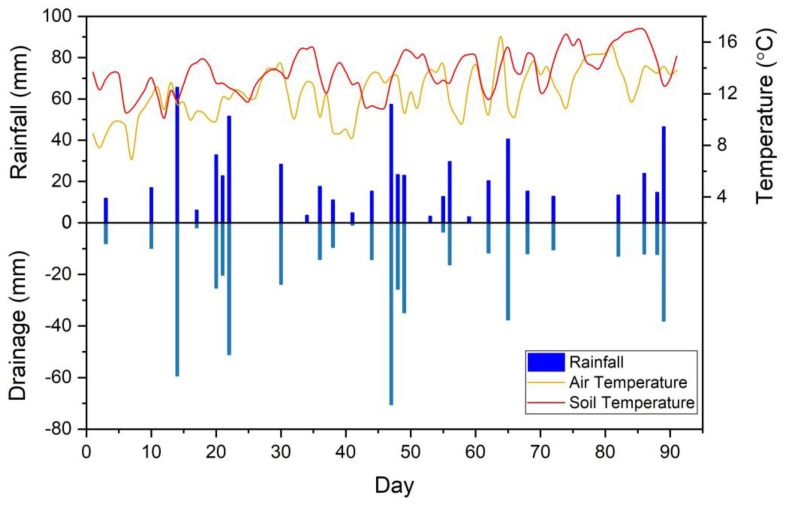
The climate data for the experimental period.

**Figure 2 plants-11-03288-f002:**
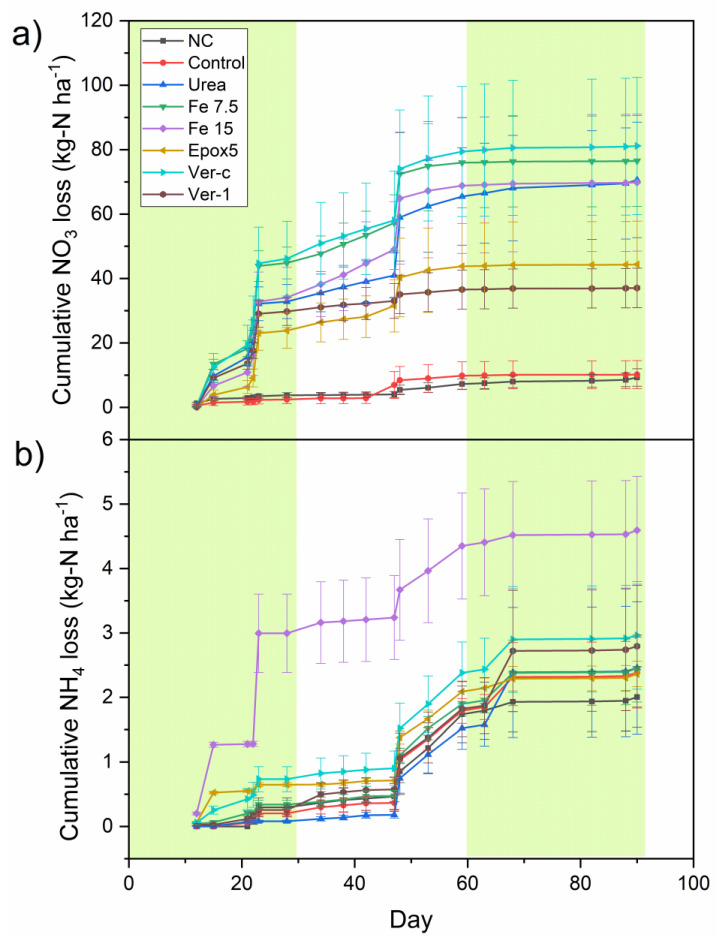
The cumulative (**a**) nitrate and (**b**) ammonium ion leaching losses during the experimental period. The error bars represent the standard error of five replicates. The green shades are used to demarcate monthly changes over the three months.

**Figure 3 plants-11-03288-f003:**
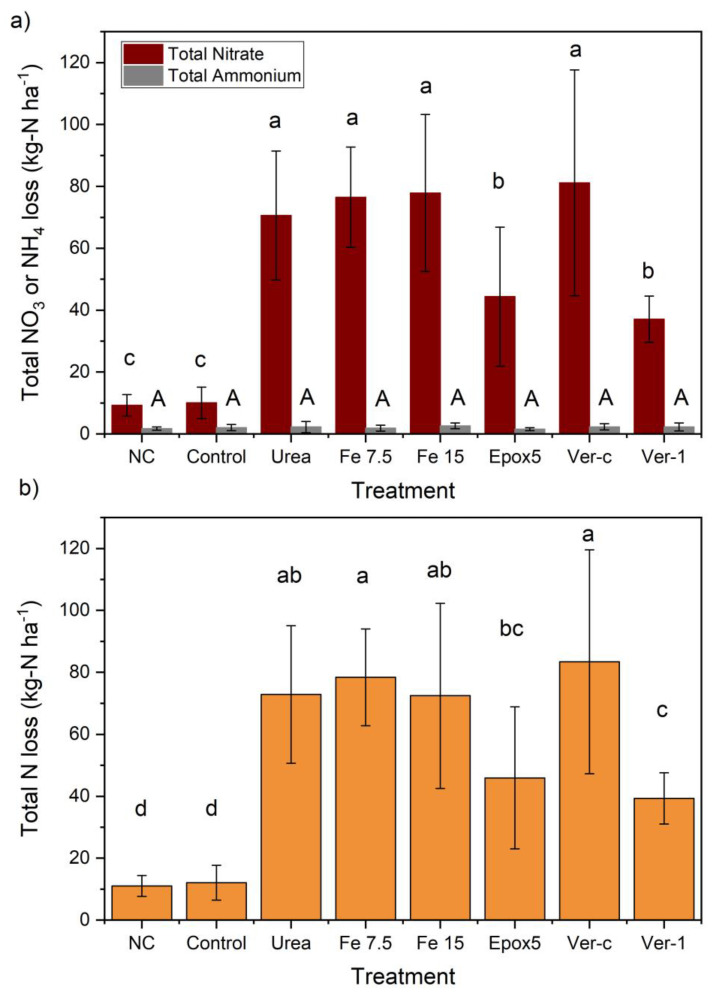
(**a**) Total nitrate and ammonium losses and (**b**) total N losses of different treatments. Error bars indicate the standard deviation (*n* = 5). The letters show a significant difference between the treatments at *p* < 0.05.

**Figure 4 plants-11-03288-f004:**
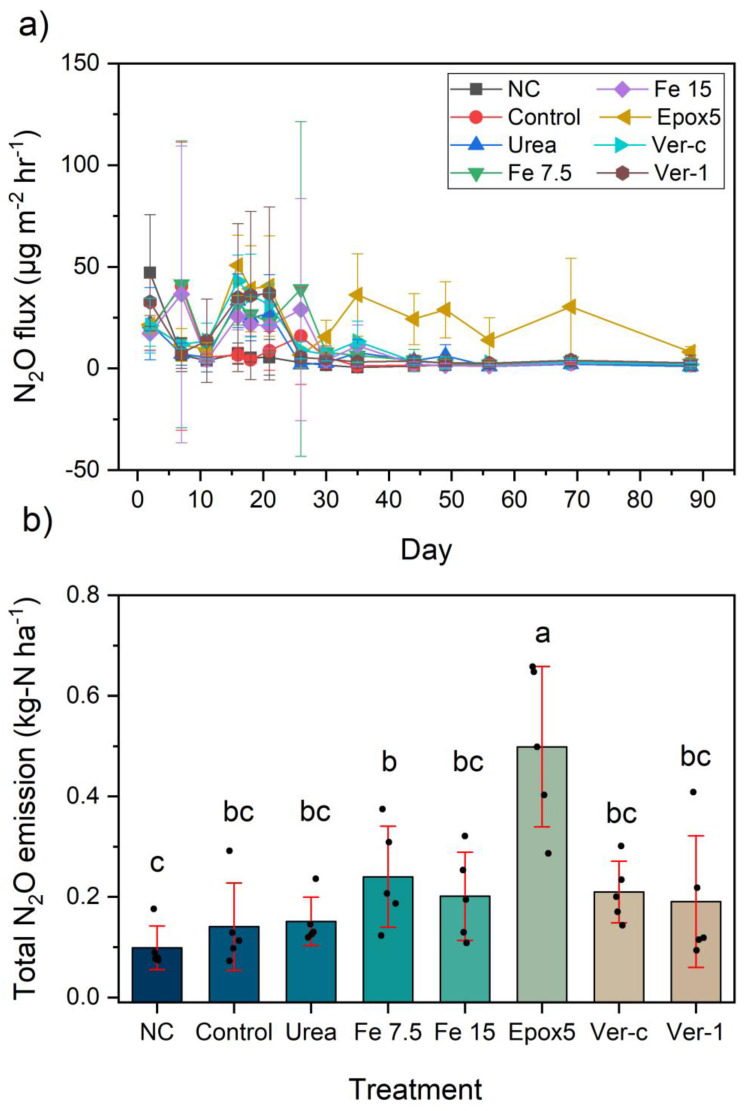
(**a**) The temporal variation of the N_2_O flux and (**b**) the bar chart with individual points comparing the total N_2_O emissions between treatments (error bars show the standard deviation and letters show a significant difference at *p* < 0.05 (*n* = 5)).

**Figure 5 plants-11-03288-f005:**
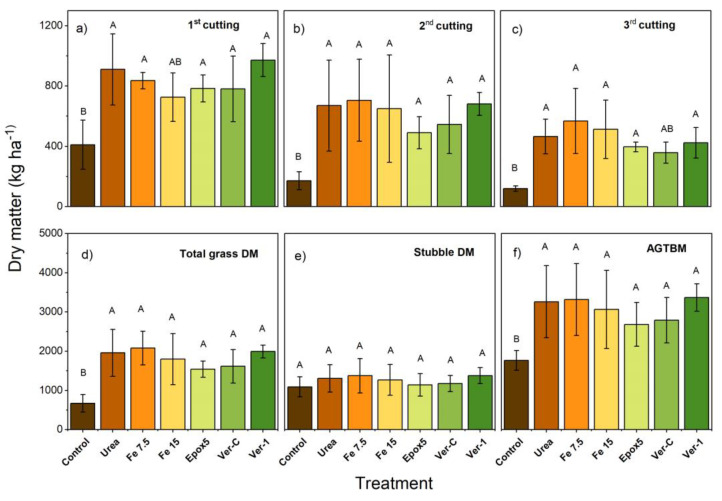
(**a**–**c**) Monthly and (**d**) total grass DM, (**e**) stubble DM and (**f**) above-ground total biomass (AGTBM) DM. The upper-case letters show the significant difference between treatments at *p* < 0.05 (*n* = 5). The error bar indicates the standard deviation (*n* = 5).

**Figure 6 plants-11-03288-f006:**
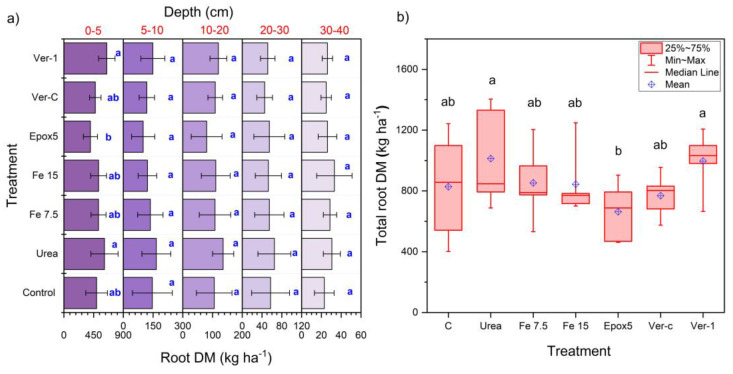
(**a**) Depth-wise root DM distribution; error bars show the standard deviation (*n* = 5) and (**b**) total root DM (TRDM) for different treatments. Different letters indicate the significant differences between the treatment within a depth in (**a**), and the total root DM in (**b**), at *p* < 0.05 (*n* = 5).

**Figure 7 plants-11-03288-f007:**
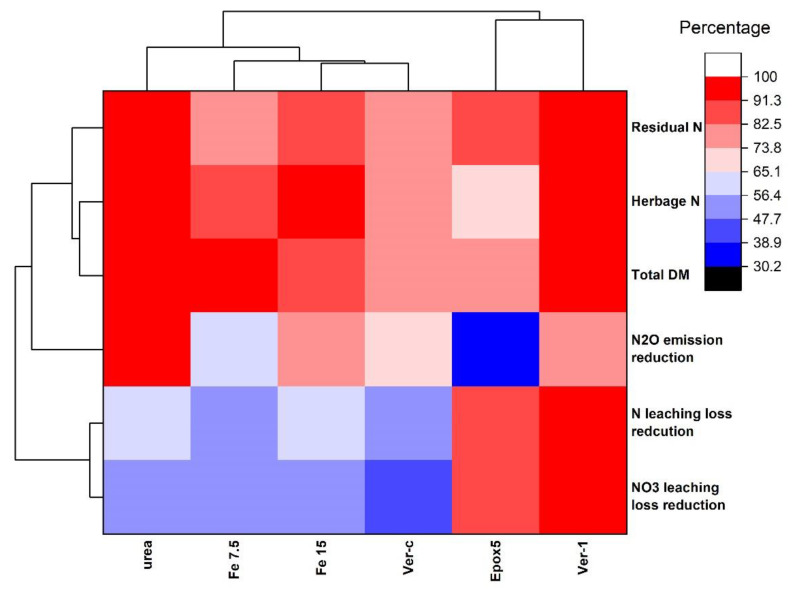
The hierarchical clustering heatmap for the classification of N treatments, based on the nitrate leaching loss reduction, total N leaching loss reduction, total DM yield, herbage N, and soil residual N. The values of all variables are in percentage (not absolute values).

**Figure 8 plants-11-03288-f008:**
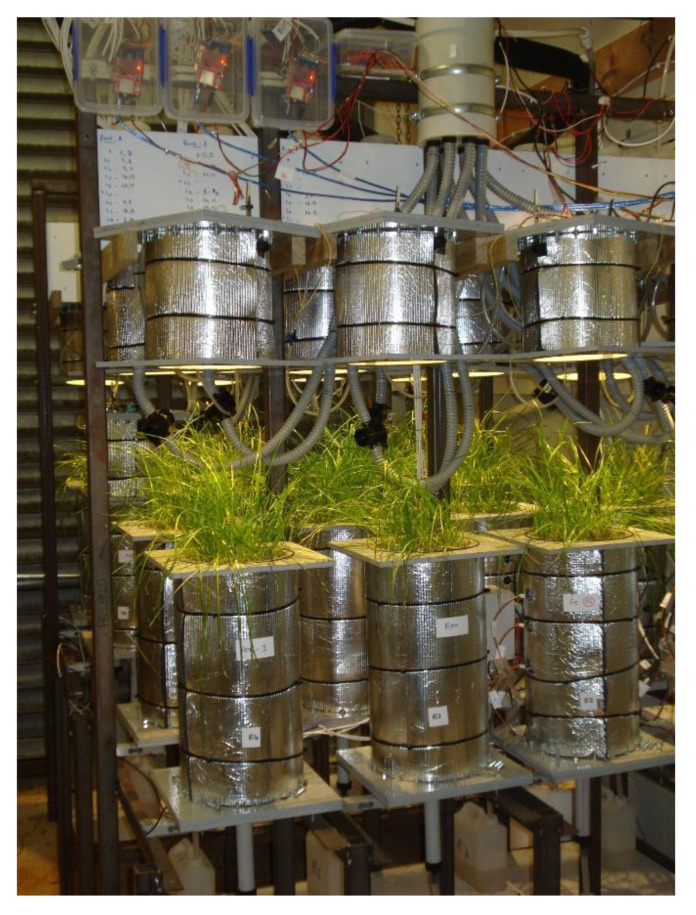
Image of the specially designed lysimeter with a cap for the climate-controlled study (This image was taken before grass clipping and caps were removed from the lysimeter body to perform grass clipping).

**Table 1 plants-11-03288-t001:** The residual mineral nitrogen in the soil profile at different depths.

Parameter	Depth (cm)	Treatment							
		NC	Control	Urea	Fe 7.5	Fe 15	Epox5	Ver-c	Ver-1
		kg-N ha^−^							
NO_3_-N	0–5	28.5 ± 6.7 ^a^	33.7 ± 10.7 ^a^	43.2 ± 13.7 ^a^	36.2 ± 11.6 ^a^	39.3 ± 14.1 ^a^	34.0 ± 23.8 ^a^	36.3 ± 12.6 ^a^	38.8 ± 23.1 ^a^
	5–10	21.1 ± 5.4 ^a^	32.7 ± 7.9 ^a^	31.7 ± 7.3 ^a^	29.5 ± 9.4 ^a^	34.5 ± 4.2 ^a^	32.6 ± 11.2 ^a^	33.4 ± 8.7 ^a^	33.0 ± 7.3 ^a^
	10–20	1.4 ± 0.6 ^a^	2.9 ± 0.7 ^a^	3.3 ± 1.1 ^a^	2.8 ± 0.4 ^a^	3.4 ± 1.8 ^a^	2.7 ± 0.4 ^a^	2.9 ± 0.9 ^a^	3.5 ± 1.4 ^a^
	20–30	1.7 ± 0.5 ^a^	2.2 ± 0.9 ^a^	2.1 ± 0.9 ^a^	1.8 ± 0.8 ^a^	2.1 ± 0.7 ^a^	2.1 ± 1.3 ^a^	1.9 ± 1.1 ^a^	2.5 ± 1.3 ^a^
	30–40	0.8 ± 0.3 ^a^	1.8 ± 1.0 ^a^	2.2 ± 1.7 ^a^	2.4 ± 1.7 ^a^	2.3 ± 1.7 ^a^	1.9 ± 1.2 ^a^	1.4 ± 0.6 ^a^	1.5 ± 0.9 ^a^
	**Total**	**53.5 ± 12.4 ^a^**	**73.2 ± 18.5 ^a^**	**82.5 ± 20.6 ^a^**	**72.7 ± 21.7 ^a^**	**81.5 ± 20.4 ^a^**	**80.3 ± 24.3 ^a^**	**76.0 ± 20.7 ^a^**	**79.3 ± 30.5 ^a^**
NH_4_-N	0–5	2.4 ± 1.2 ^a^	3.8 ± 2.3 ^a^	3.9 ± 2.2 ^a^	3.2 ± 1.7 ^a^	3.3 ± 1.2 ^a^	3.0 ± 3.0 ^a^	3.1 ± 1.9 ^a^	3.8 ± 5.5 ^a^
	5–10	2.1 ± 0.9 ^a^	2.5 ± 1.1 ^a^	2.8 ± 1.1 ^a^	2.7 ± 1.5 ^a^	3.1 ± 1.5 ^a^	2.3 ± 1.2 ^a^	2.2 ± 1.5 ^a^	2.6 ± 2.3 ^a^
	10–20	1.1 ± 1.1 ^a^	1.9 ± 1.2 ^a^	1.3 ± 1.8 ^a^	0.9 ± 1.0 ^a^	1.9 ± 1.8 ^a^	2.2 ± 2.3 ^a^	1.6 ± 2.4 ^a^	1.5 ± 1.5 ^a^
	20–30	0.5 ± 0.8 ^a^	1.8 ± 1.6 ^a^	1.3 ± 2.0 ^a^	0.2 ± 0.4 ^a^	1.4 ± 1.4 ^a^	1.0 ± 1.0 ^a^	0.7 ± 1.3 ^a^	0.9 ± 0.9 ^a^
	30–40	0.4 ± 0.7 ^a^	1.9 ± 1.7 ^a^	1.7 ± 2.2 ^a^	1.1 ± 1.4 ^a^	1.4 ± 1.7 ^a^	1.1 ± 1.0 ^a^	0.5 ± 1.0 ^a^	1.2 ± 1.9 ^a^
	**Total**	**6.5 ± 2.8 ^a^**	**14.4 ± 1.7 ^a^**	**13.3 ± 6.3 ^a^**	**6.5 ± 3.3 ^a^**	**11.1 ± 6.7 ^a^**	**11.5 ± 5.9 ^a^**	**9.4 ± 7.9 ^a^**	**12.0 ± 9.6 ^a^**
Total-N	0–5	30.9 ± 8.5 ^a^	37.5 ± 12.2 ^a^	47.0 ± 15.7 ^a^	39.4 ± 12.2 ^a^	42.6 ± 15.1 ^a^	46.3 ± 19.2 ^a^	39.4 ± 14.4 ^a^	42.6 ± 28.2 ^a^
	5–10	23.2 ± 5.8 ^a^	35.2 ± 8.8 ^a^	34.5 ± 8.1 ^a^	32.3 ± 10.8 ^a^	37.6 ± 4.7 ^a^	34.9 ± 11.8 ^a^	35.6 ± 9.4 ^a^	35.6 ± 8.6 ^a^
	10–20	2.5 ± 1.6 ^a^	4.8 ± 1.8 ^a^	4.6 ± 2.6 ^a^	3.6 ± 0.7 ^a^	5.3 ± 3.3 ^a^	5.0 ± 2.5 ^a^	4.5 ± 2.5 ^a^	4.9 ± 2.6 ^a^
	20–30	2.2 ± 1.3 ^a^	3.9 ± 2.3 ^a^	3.4 ± 2.9 ^a^	1.9 ± 1.1 ^a^	3.4 ± 2.0 ^a^	3.1 ± 2.0 ^a^	2.6 ± 2.4 ^a^	3.3 ± 1.5 ^a^
	30–40	1.2 ± 1.5 ^a^	3.7 ± 2.6 ^a^	3.9 ± 3.8 ^a^	3.5 ± 3.1 ^a^	3.7 ± 3.4 ^a^	3.0 ± 2.1 ^a^	1.9 ± 1.6 ^a^	2.7 ± 2.3 ^a^
	**Total**	**60.0 ± 14.3 ^a^**	**93.2 ± 16.8 ^a^**	**104.4 ± 8.9 ^a^**	**77.1 ± 27.2 ^a^**	**92.6 ± 25.6 ^a^**	**91.8 ± 28.1 ^a^**	**84.1 ± 26.6 ^a^**	**100.5 ± 31.7^a^**

**Data presented as mean ± standard deviation of five replicates.** Mean values with the same lower-case letters within a row are not significantly different between treatments (*p* < 0.05).

**Table 2 plants-11-03288-t002:** The herbage nitrogen and nitrogen utilisation efficiencies of the different treatments.

Treatment	TN in 1st Cutting	TN in 2nd Cutting	TN in 3rd Cutting	TN in Stubble	Total Herbage N	AUE	ARE
	(kg-N ha^−1^)					(kg-DM kg-N^−1^)	-
Control	10.8 ± 4.7 ^d^	3.3 ± 1.1 ^c^	1.5 ± 1.1 ^b^	9.5 ± 3.7 ^b^	15.6 ± 6.2 ^b^	-	-
Urea	44.1 ± 12.4 ^ab^	23.0 ± 10.7 ^ab^	6.4 ± 3.6 ^a^	18.0 ± 2.8 ^a^	73.5 ± 25.8 ^a^	6.4 ± 3.0 ^a^	0.29 ± 0.13 ^a^
Fe 7.5	35.9 ± 2.7 ^abc^	22.1 ± 8.7 ^ab^	6.6 ± 2.1 ^a^	15.3 ± 7.7 ^ab^	66.7 ± 5.5 ^a^	7.1 ± 2.1 ^a^	0.26 ± 0.03 ^a^
Fe 15	34.0 ± 8.2 ^bc^	29.7 ± 9.2 ^a^	10.0 ± 2.0 ^a^	16.1 ± 2.7 ^ab^	77.3 ± 20.1 ^a^	5.6 ± 3.2 ^a^	0.31 ± 0.10 ^a^
Epox5	33.8 ± 6.3 ^c^	17.9 ± 3.2 ^b^	7.6 ± 3.9 ^a^	17.8 ± 6.9 ^a^	56.7 ± 12.5 ^a^	5.0 ± 1.0 ^a^	0.21 ± 0.06 ^a^
Ver-C	38.2 ± 11.5 ^abc^	18.3 ± 6.0 ^b^	5.8 ± 2.6 ^a^	17.8 ± 2.8 ^a^	62.3 ± 13.8 ^a^	4.7 ± 2.1 ^a^	0.23 ± 0.07 ^a^
Ver-1	44.8 ± 3.7 ^a^	20.7 ± 3.3 ^ab^	8.5 ± 3.3 ^a^	17.5 ± 6.6 ^a^	77.1 ± 2.5 ^a^	6.6 ± 0.8 ^a^	0.30 ± 0.03 ^a^

Data are mean ± standard deviation error of five replicates. Values followed by different lower-case letters within a column for each treatment are significantly different at *p* < 0.05. TN, AUE and ARE refer to the total N, agronomic use efficiency and apparent recovery efficiency, respectively. AUE and ARE are the additional DM yield and additional N accumulation of a N treatment, compared to the control for a unit kg-N application, respectively.

**Table 3 plants-11-03288-t003:** The N balance of different treatments.

Variable	Treatment							
	NC	Control	Urea	Fe 7.5	Fe 15	Epox5	Ver-c	Ver-1
	kg-N ha^−1^							
Fertiliser N	0	0	200	200	200	200	200	200
Mineralised N	84.6	84.6	84.6	84.6	84.6	84.6	84.6	84.6
SIMN *	46	46	46	46	46	46	46	46
Herbage N	-	−15.6	−73.5	−66.7	−77.3	−56.7	−62.3	−77.1
Stubble N	-	−9.5	−18	−15.3	−16.1	−17.8	−17.8	−17.5
Leaching N	−11.0	−12.1	−72.9	−78.4	−80.5	−45.9	−83.4	−39.3
N_2_O-N	−0.1	−0.1	−0.2	−0.2	−0.2	−0.5	−0.2	−0.2
SFMN *	−60.0	−93.2	−104	−77.1	−92.6	−91.8	−84.1	−100.5
Unreleased N	0	0	0	0	0	9.7	0	0
N Balance	59.4	0.0	61.6	92.8	63.8	108.1	82.8	95.9

Inflow and outflow variables are indicated in positive and negative signs, respectively. * SIMN & SFMN stand for initial and final soil mineral nitrogen, respectively.

## Data Availability

Data will be available from the corresponding author upon reasonable request.

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
