# Peer review of "The Nitrogen Dynamics of Newly Developed Lignite-Based Controlled-Release Fertilisers in the Soil-Plant Cycle"

_plants, 2022, doi:10.3390/plants11233288_

Round 1
Reviewer 1 Report
Check English again
Include more relevant citation in introduction, 13 is not sufficient
Better to share statistical analysis with me
give percent increase of the recommended treatment values in comparison to other treatments in both abstract and conclusions
Author Response
Responses for reviewer 1
Manuscript number: Plants-2031622
Title of the manuscript: The N nitrogen dynamics of newly developed lignite-based controlled-release fertilisers in the soil-plant cycle
We are grateful to the reviewer for the constructive comments. We have addressed the comments carefully which substantially improved our manuscript. Our response to each of the comments from the reviewer is presented here in BLUE. Line numbers refer to the new manuscript.
Reviewer comments:
- Reviewer comment 1: Check English again
Author response:
Thank for your comment. The English is checked by native speakers.
- Reviewer comment 2: Include more relevant citation in introduction, 13 is not sufficient.
Author response:
We appreciate the reviewer’s comment and more citations are added.
- Reviewer comment 3: Better to share statistical analysis with me.
Author response:
The statistical output is attached as a separate document.
- Reviewer comment 4: give percent increase of the recommended treatment values in comparison to other treatments in both abstract and conclusions
Author response:
We are grateful for this comment. The percentage of treatment effects are added to the abstract and conclusion as suggested. Please see lines 26 and 534.
Reviewer 2 Report
The article is not innovative, but it has a good role in promoting production practice. Not refined enough in results and analysis. Iron is very important for DIR process, but there is a lack of relevant data in the article. There are some mistakes in grammar, unit and abbreviation in the article, which need to be further revised. On the whole, my suggestion is to make a major revision before considering whether to publish it.
Abstract
The nitrogen in abstract has no abbreviation, so it's better to unify it. When the first occurrence of nitrogen is marked with abbreviations, the following ones are replaced by abbreviations.
Introduction
Line 39 [3] ,The reference is cited incorrectly.
Line 52 Nitrogen needs abbreviations. Similar errors need to be corrected in the text.
Line 71 A specially designed lysimeter system is interesting, Please provide some photos.
Line 91 50 kg-N ha-1. The unit format needs attention.
Line 105 There are many units in the text without superscript.
Line 95 Why is the water content designed to be 100% at the end of three months?
Materials and Methods
Line 97-line 112. How is the fertilizer applied, on the surface, or in the soil below a few centimeters? It was applied several times during the experiment, and details need to be given.
Line 199-line 212 In the process of nitrogen balance calculation, are other gases containing N, such as NH3,NO compounds and N2, not considered? Especially, the emission of NH3 is generally significantly larger than N2O?
Results and Discussion
Figure 2 It is suggested to increase the calculation of nitrogen loss by monthly or critical time nodes. Shown in figure 2. Or add a histogram.
Parts of Line 238-243 and line 247-255 are repeated, so it is suggested to merge them into one paragraph.
As the author said, the role of ferrous ions (Fe2+) and ferric ions (Fe3+) in nitrogen cycle is not completely clear, and is a topic studied in recent times. However, dissimilary iron-reduction (DIR) rates was not determined in this paper. There is no direct evidence to prove the effect of iron fertilizer. It is suggested to increase this indicator.
Author Response
Responses for reviewer 2
Manuscript number: Plants-2031622
Title of the manuscript: The N nitrogen dynamics of newly developed lignite-based controlled-release fertilisers in the soil-plant cycle
We are grateful to reviewer for the thoughtful comments. We have addressed the comments carefully which substantially improved our manuscript. Our response to each of the comments from the reviewer is presented here in BLUE. Line numbers refer to the new manuscript.
Reviewer comments:
- Reviewer comment 1: The nitrogen in the abstract has no abbreviation, so it's better to unify it. When the first occurrence of nitrogen is marked with abbreviations, the following ones are replaced by abbreviations.
Author response:
We appreciate the reviewer’s comment to add the abbreviation for nitrogen. The suggested amendment was made in the revised manuscript. Please see line 16.
- Reviewer comment 2: Line 39 [3], The reference is cited incorrectly.
Author response:
Thank you for highlighting the error in the citation. It is changed as “Richards, et al. [3]”. Please see line 39.
- Reviewer comment 3: Line 71 A specially designed lysimeter system is interesting, please provide some photos.
Author response:
Thank you for your encouragement and interest in the image in our newly developed lysimeter system. An image of the lysimeter is added (Figure 1) which shows the lysimeter with ryegrass before grass clipping. Therefore, the caps are removed from the lysimeter body and placed in the cap holder to facilitate the grass clipping.
- Reviewer comment 4: Line 91: 50 kg-N ha-1.The unit format needs attention.
Author response:
Thank you for pinpointing the mistake in the unit. The part of the unit is superscripted. Please see line 95.
- Reviewer comment 5: Line 105 There are many units in the text without superscript.
Author response:
We apologise for the mistake in the units. The document is now carefully corrected for the units.
- Reviewer comment 6: Line 95 Why is the water content designed to be 100% at the end of three months?
Author response:
We thank the reviewer for this query. We maintained the soil moisture at 100% at the start of the experiment for two reasons;
- This method made it easy to do the water balance calculation. The water balance calculation is given below;
?=?−??−?−?
Where D is drainage, I is irrigation, ET is evapotranspiration, S is soil water storage, and R is runoff.
If the soil moisture is maintained at 100%, the change of the soil water storage (S) becomes zero. Therefore, this component can be eliminated from the equation.
- The soil moisture across all experimental units is uniform and therefore, it doesn’t impact the results of this study.
We understand that we better include this in the manuscript. Therefore, we incorporated the following statement within the text; “to maintain a uniform moisture content for all experimental units and facilitate the water balance calculations”. Please see lines 99-101.
- Reviewer comment 7: Line 97-line 112.How is the fertilizer applied, on the surface, or in the soil below a few centimetres? It was applied several times during the experiment, and details need to be given.
Author response:
We are grateful for this comment which we have taken into consideration to improve our manuscript. The fertiliser application details are now included. Please see lines 124-125.
- Reviewer comment 8: Line 199-line 212 In the process of nitrogen balance calculation, are other gases containing N, such as NH3, NO compounds and N2, not considered? Especially, the emission of NH3 is generally significantly larger than N2O?
Author response:
We appreciate the reviewer’s question on other N contacting gases in N balance calculations. One of the major focuses of the present study is to minimise the N leaching losses and nitrous oxide (N2O) emissions. NH3 is a short-living species in the environment and therefore, the impact of NH3 in the greenhouse effect is minimal. In addition, the higher moisture content of soil due to high rainfall pattern in this study doesn’t favour of ammonia release. For these reasons, NH3 concentration was not measured and not included in the N balance calculation.
We added a sentence in the manuscript to clarify this; “The prime interest of this study is to minimise the N leaching losses and nitrous oxide (N2O) emissions via developing CRFs and therefore, other N gas species were not measured and considered in N balance”. Please see lines 227-229.
- Reviewer comment 9: Figure 2 It is suggested to increase the calculation of nitrogen loss by monthly or critical time nodes. Shown in figure 2. Or add a histogram.
Author response:
We are grateful for this comment. Colour shades are now added to highlight three months which enables the reader to understand the cumulative nitrate and ammonium losses in all three months. Please see Figure 3.
- Reviewer comment 10: Parts of Line 238-243 and line 247-255 are repeated, so it is suggested to merge them into one paragraph.
Author response:
Thank you for your comments. The sentences were rearranged as suggested. Please see lines 264-268.
- Reviewer comment 11: As the author said, the role of ferrous ions (Fe2+) and ferric ions (Fe3+) in nitrogen cycle is not completely clear, and is a topic studied in recent times. However, dissimilary iron-reduction (DIR) rates was not determined in this paper. There is no direct evidence to prove the effect of iron fertilizer. It is suggested to increase this indicator.
Author response:
We appreciate the reviewer's comment to highlight the DIR process in the manuscript. The DIR rates can only be measured in a controlled incubation study and are not possible with a lysimeter study. As suggested, a discussion about DIR pathway is added in the conclusion. Please see lines 540-543.
Round 2
Reviewer 2 Report
The author revised it according to my suggestions, and I think it is still satisfactory, although the whole article is generally innovative. My general suggestion is that this article can be accepted for publication.